# Who Is Afraid of CRP? Elevated Preoperative CRP Levels Might Attenuate the Increase in Inflammatory Parameters in Response to Lung Cancer Surgery

**DOI:** 10.3390/jcm9103340

**Published:** 2020-10-18

**Authors:** Moritz Mecki Meyer, Leon Brandenburg, Helge Hudel, Alisa Agné, Winfried Padberg, Ali Erdogan, Holger Nef, Anca-Laura Amati, Oliver Dörr, Biruta Witte, Veronika Grau

**Affiliations:** 1Laboratory of Experimental Surgery, German Center for Lung Research, Department of General and Thoracic Surgery, Justus-Liebig-University Giessen, D-35385 Giessen, Germany; Moritz.M.Meyer@chiru.med.uni-giessen.de (M.M.M.); Leon.Brandenburg@chiru.med.uni-giessen.de (L.B.); Alisa.Agne@chiru.med.uni-giessen.de (A.A.); Winfried.Padberg@chiru.med.uni-giessen.de (W.P.); Anca-Laura.Amati@chiru.med.uni-giessen.de (A.-L.A.); Biruta.Witte@chiru.med.uni-giessen.de (B.W.); 2Institute of Medical Informatics Medical Statistics, Justus-Liebig-University Giessen, D-35385Giessen, Germany; Helge.Hudel@informatik.med.uni-giessen.de; 3Internist Practice Center, Balserische Stiftung Hospital, D-35392 Giessen, Germany; erdogan@ipz-giessen.de; 4German Centre for Cardiovascular Research (DZHK), Department of Cardiology, University of Giessen, D-35385 Giessen, Germany; Holger.Nef@innere.med.uni-giessen.de (H.N.); oliver.doerr@innere.med.uni-giessen.de (O.D.)

**Keywords:** atorvastatin, C-reactive protein, fever, gender, inflammatory complications, interleukin-1β, leukocyte count, lung cancer, sepsis, thoracic surgery

## Abstract

During surgery, ATP from damaged cells induces the release of interleukin-1β, a potent pro-inflammatory cytokine that contributes to the development of postoperative systemic inflammation, sepsis and multi-organ damage. We recently demonstrated that C-reactive protein (CRP) inhibits the ATP-induced release of monocytic interleukin-1β, although high CRP levels are deemed to be a poor prognostic marker. Here, we retrospectively investigated if preoperative CRP levels correlate with postoperative CRP, leukocyte counts and fever in the context of anatomical lung resection and systematic lymph node dissection as first line lung cancer therapy. No correlation was found in the overall results. In men, however, preoperative CRP and leukocyte counts positively correlated on postoperative days one to two, and a negative correlation of CRP and fever was seen in women. These correlations were more pronounced in men taking statins and in statin-naïve women. Accordingly, the inhibitory effect of CRP on the ATP-induced interleukin-1β release was blunted in monocytes from coronary heart disease patients treated with atorvastatin compared to monocytes obtained before medication. Hence, the common notion that elevated CRP levels predict more severe postoperative inflammation should be questioned. We rather hypothesize that in women and statin-naïve patients, high CRP levels attenuate trauma-induced increases in inflammatory markers.

## 1. Introduction

Surgical procedures are inevitably associated with systemic inflammation, the degree of which depends on the severity of the associated tissue damage [1,2]. Inflammation is caused by numerous damage-associated molecular patterns (DAMPs) that are liberated from damaged cells and extracellular matrices. DAMPs induce the synthesis and release of a multitude of pro-inflammatory mediators that play essential roles in wound healing and in the elimination of debris or invading pathogens [1,2]. The other side of the coin, however, is the risk of exaggerated inflammation that can result in systemic inflammatory response syndrome (SIRS). SIRS is a life-threatening condition that may further develop towards sepsis and multi-organ dysfunction [1,2,3,4]. A prominent example of these Janus-faced pro-inflammatory cytokines is interleukin-1β (IL-1β) which is produced by mononuclear phagocytes and a range of other cells [1,2,5,6].

The production and secretion of IL-1β is tightly controlled and typically requires two independent danger signals [4,7,8,9,10,11]. Signal one may be DAMPs or pathogen-associated molecular patterns (PAMPs) that induce synthesis of the biologically inactive pro-form of IL-1β and of components of the NACHT, LRR and PYD domains-containing protein 3 (NLRP3) inflammasome. In the context of surgical trauma, extracellular ATP originating from the cytoplasm of damaged cells is a typical second danger signal that activates the P2X7 receptor, induces the assembly of the NLRP3 inflammasome and activates caspase-1 [1]. The protease caspase-1 cleaves pro-IL-1β and mature IL-1β is formed and swiftly released [4,7,8,9,10,11].

C-reactive protein (CRP) is an acute-phase protein, frequently used as a sensitive clinical indicator of systemic inflammation [12,13,14]. CRP is mainly produced in the liver in response to IL-1β and IL-6. Disk-shaped CRP homopentamers are secreted into the circulation [12,13,14]. Each CRP subunit has the capability to bind one phosphocholine head-group in a Ca^2+^-dependent way [12,13,14]. Upon binding to phosphocholine head-groups on the surface of damaged cells or pathogens, a transition towards a pro-inflammatory conformation of CRP is initiated. This transition unmasks binding sites for the complement protein C1q and Fc receptors and causes the formation of pro-inflammatory CRP monomers [15,16]. The transition of CRP mainly takes place in injured tissue and seems to contribute to ischemia and reperfusion injuries, including myocardial infarction [17,18,19]. By contrast, there is ample, albeit disputed, experimental evidence that CRP exerts anti-inflammatory functions [20,21,22,23].

Our laboratory discovered that typical ligands of CRP, such as dipalmitoyl phosphatidylcholine, lysophosphatidylcholine, glycerophosphocholine or phosphocholine, are potent inhibitors of the ATP-induced release of IL-1β by lipopolysaccharide (LPS)-primed human monocytes [24,25,26,27,28]. These phosphocholines stimulate monocytic nicotinic acetylcholine receptors that efficiently inhibit the activation of P2X7 receptors and, hence, inhibit the release of bioactive IL-1β. Recently, we demonstrated that the anti-inflammatory activity of phosphocholine is markedly enhanced by its interaction with CRP [29]. We further provided clinical data suggesting that CRP functions as a negative feedback regulator of IL-1β release in the context of multiple traumata [29]. However, more studies are warranted to define the role of CRP in the context of surgical trauma.

In this study, we investigate if high preoperative CRP levels correlate with postoperative hallmarks of inflammation in patients undergoing pulmonary resection for the treatment of lung cancer. We performed a single-center, retrospective pilot study on 217 patients who underwent resection of primary lung tumors. Patients were stratified according to the type of surgery, smoking behavior, gender and lipid-lowering therapy. Correlation analyses were performed that compare preoperative CRP concentrations to postoperative CRP levels, body temperature and leukocyte counts. In a second part of the study, the effect of lipid-lowering medication on the CRP-mediated control of monocytic IL-1β release was studied ex vivo. Our results suggest that high preoperative CRP levels do not predict more severe postoperative inflammation. CRP may, rather, have an anti-inflammatory potential in women and in the absence of lipid-lowering therapy.

## 2. Materials and Methods

### 2.1. Lung Cancer Patients

The retrospective analysis of data from patients who underwent thoracic surgery was approved by the local ethics committee (medical faculty, University of Giessen, Germany, approval No. 104/17), who waived the need for informed consent of the patients involved because of the retrospective design of the study and the lack of direct interventions.

Eligible patients of the University Hospital Giessen, Germany, who underwent curative lung cancer surgery, were included into this retrospective cohort study between 1 April 2012 and 31 March 2017. Patients were excluded from the study when the preoperative CRP level was not documented; when the surgical procedure was abandoned; in the case of acute inflammatory thoracic disease, acute infection, severe inflammatory disease, preoperative leukopenia (≤3500/µL), preoperative leukocytosis (≥12,000/µL), preoperative fever (≥38 °C), systemic treatment with corticoids, therapeutic application of antibodies, chemo- or radiotherapy or a history of transplantation. Tumors were staged according to the seventh UICC (union internationale contre le cancer) guidelines [30,31,32,33]. Blood samples for leukocyte count and CRP measurement were obtained routinely within 7 days prior to surgery and on postoperative day 1. Thereafter, blood sampling frequency was at the discretion of the treating physician, dependent on the clinical course and ended with discharge from the hospital. Temperature was taken every morning by the means of an electronic auricular thermometer.

Subgroups were formed for further analyses regarding gender (male or female), smoking habit (smoker, ex-smoker or unknown smoking habit), surgical access (video-assisted thoracic surgery (VATS), conversion from VATS to thoracotomy or thoracotomy), extent of pulmonary resection (segmentectomy, lobectomy or pneumonectomy) and postoperative inflammatory complications.

### 2.2. Coronary Heart Disease Patients

Patients were recruited upon informed written consent among patients, who were admitted to our hospital for coronary angiography. The local ethics committee approved this study (medical faculty, University of Giessen, Germany, approval No. 35/19). Inclusion criteria were an age above 18 years and a low-grade coronary heart disease requiring prescription of lipid-lowering medicaments. Exclusion criteria were acute coronary syndrome, limb ischemia, infections, chronic renal insufficiency, pregnancy or use of statins, anticoagulants, antibiotics or immunosuppressants. Blood was taken shortly (i) before coronary angiography, (ii) after four weeks of daily medication with 40 mg atorvastatin (Sortis^®^, Pfizer Pharma, Berlin, Germany) and (iii) after another 4 weeks of atorvastatin in combination with 100 mg acetylsalicylic acid (ASA, Aspirin^®^, Bayer, Leverkusen, Germany) per day.

### 2.3. IL-1β Release Experiments

Monocytes were enriched from heparinized patient blood (18.75 IU heparin/mL, heparin sodium, Ratiopharm GmbH, Ulm, Germany) using the RosetteSep^TM^ human Monocyte Enrichment Cocktail (Stemcell Technologies, Köln, Germany). Cells were seeded in 96-well microtiter plates at a concentration of 1 × 10^5^ cells/250 µL in the cell culture medium (RPMI) (Roswell Park Memorial Institute) 1640 (Sigma-Aldrich via Merck KGaA, Darmstadt, Germany) and incubated (37 °C, 5% CO_2_) for 3 h in the presence or absence of lipopolysaccharide (LPS, 5 ng/mL, *Escherichia coli*, serotype O26:B6, Sigma-Aldrich). Thereafter, the cells were stimulated with the P2X7 receptor agonist 2′(3′)-*O*-(4-benzoylbenzoyl)adenosine 5′-triphosphate triethylammonium salt (BzATP, 100 µM) in the presence or absence of acetylcholine (ACh, 100 µM, Sigma-Aldrich), phosphocholine (PC, 200 µM, Sigma-Aldrich), CRP isolated from human pleural fluid (5 µg/mL, Merck, Darmstadt, Germany), nicotine hydrogen tartrate (100 µM, Sigma-Aldrich), l-α-glycerophosphocholine (GPC, 100 µM, Sigma-Aldrich) or dipalmitoyl phosphatidylcholine (DPPC, 100 µM, Sigma-Aldrich). After another 30 min of incubation, cell-free cell culture supernatants were harvested and stored at −20 °C until measurement of IL-1β concentrations and lactate dehydrogenase (LDH) activity.

### 2.4. Blood Analyses

CRP, cholesterol, low density lipoprotein (LDL)-cholesterol, high density lipoprotein (HDL)-cholesterol and triglycerides were analyzed in patient blood serum by the Institute of Laboratory Medicine at the University Hospital Giessen or by the Bioscientia Labor Mittelhessen, both in Giessen, Germany.

### 2.5. Measurement of IL-1β Concentrations

IL-1β concentrations in the cell-free cell culture supernatants were measured using the Human IL-1beta/IL-1F2 DuoSet sandwich ELISA (R&D Systems, Minneapolis, MN, USA) according to the instructions of the supplier.

### 2.6. Estimation of Cell Death

To estimate cell death, the Non-Radioactive Cytotoxicity Assay (Promega, Madison, WI, USA) was used to measure LDH activity as indicated by the instructions of the provider. LDH activity is given as percentage of the total LDH activity released by lyzed untreated control cells. Cell viability was not impaired under the above described experimental conditions.

### 2.7. Statistical Evaluation

SPPS^®^ (Versions 23-25, IBM^®^, Armonk, NY, USA) was used for statistical analyses of all data. Data sets were tested for normal distribution using the Shapiro–Wilk-Test. As CRP values were not normally distributed, non-parametric tests were used throughout. To analyze the relationships between preoperative CRP levels and postoperative CRP levels, leukocyte counts or body temperature Spearman’s rank correlation coefficients (*r*) were determined. Data are visualized as scatterplots, including a best fit straight line. As the retrospective part of this study that investigates patients undergoing pulmonary resection is an exploratory pilot study obscured by multiple testing, we have to point out that this part of the study only allows for the formation of hypotheses. Data obtained by the IL-1β release assays were analyzed by the Friedman test, followed by the Wilcoxon signed-rank test. Here, *p* ≤ 0.05 is considered to be statistically significant.

## 3. Results

### 3.1. CRP and Inflammation in Response to Surgical Extirpation of Primary Lung Tumors

#### 3.1.1. Patient Characteristics

In total, 301 patients who underwent anatomical lung resection and systematic lymph node dissection as first-line lung cancer therapy were considered for inclusion in this retrospective study (Figure 1). Among them, 84 patients were excluded: 16 patients suffered from preoperative inflammatory disease, 52 were treated with immunomodulatory medication, 14 patients had preoperative leukocytosis, one had fever above 38 °C and preoperative CRP levels were not documented for one patient. The characteristics of the remaining 217 patients are summarized in Table 1. Patients with higher tumor stages typically presented with higher preoperative CRP levels (Appendix A). Only 2.8% of all patients included were younger than 50 years and 4.6% were older than 80 years. Preoperative CRP levels were lower in female patients and the proportion of carcinoid tumors differed among male and female patients. Data of these patients were analyzed in an exploratory manner.

#### 3.1.2. Total Patient Population

As expected, CRP levels, leukocyte counts and body temperature increased in the early postoperative phase (Appendix A). Linear regression analysis of preoperative versus postoperative CRP values revealed a positive correlation at all postoperative days (days 1 to 10), and correlation coefficients continuously dropped (Table 2, Figure 2a,b). In contrast, preoperative CRP levels and postoperative leukocyte numbers did not correlate on days 1 to 2 and 6 to 10; only a weak positive correlation was seen on days 3 to 5 after surgery (Table 2; Figure 2c,d). Body temperature did not correlate at all with preoperative CRP values (Table 2; Figure 2e,f). Of note, only few patients suffered from body temperatures below 36 °C throughout the study (Appendix A).

#### 3.1.3. Smoking Behavior

Because the CRP-induced signaling cascade controlling ATP-mediated IL-1β release by human monocytic cells involves nicotinic acetylcholine receptors [29], and the expression of these receptors might be changed by smoking [34], patients were divided into three subgroups depending on their smoking behavior (own statements): smokers, ex-smokers, who quit smoking at least two months before surgery, and a poorly defined group with unknown smoking behavior. The latter contains all non-smokers but we cannot exclude that this group also contains smokers and ex-smokers. In the patient group with unknown smoking habits, preoperative CRP values positively correlated with postoperative CRP values at all postoperative days except on day 8 (Appendix A). This positive correlation was partially lost in the smoking subgroup on postoperative days 2 to 4 and 7 to 10 as well as in ex-smokers from days 4 to 10, which might be due to small patient number in the subgroups (Appendix A). Regarding leukocytes and fever, a positive correlation of preoperative CRP and leukocyte counts on day two in ex-smokers (*r* = 0.612, *n* = 18, *p* = 0.007), was the only difference to the total patient population (Table 2, Appendix A).

#### 3.1.4. Surgical Access and Extend of Pulmonary Resection

ATP released from damaged cells during surgery is thought to be an important trigger of early postoperative inflammation. We stratified patients according to surgical access and the extent of pulmonary resection in an attempt to investigate the differential effect of more or less trauma. In most patients, the surgical access was VATS (*n* = 152) and the type of pulmonary resection was lobectomy (*n* = 195). The results of the correlation analyses in the subgroup of VATS patients (Appendix A) and lobectomy patients (Appendix A) resembled, at large, the total patient population (Figure 2, Table 2). The size of the subgroups, where VATS was converted to thoracotomy (*n* = 22) and thoracotomy alone (*n* = 43), as well as the size of the patient groups undergoing segmentectomy (*n* = 15) or pneumonectomy (*n* = 7) are small and should be interpreted with care (Appendix A). It is, however, conspicuous that in the subgroup in which VATS was converted to thoracotomy, preoperative CRP negatively correlated with body temperature on days 6 (*r* = −0.485, *n* = 19, *p* = 0.035) and 7 (*r* = −0.675, *n* = 18, *p* = 0.002) after surgery.

#### 3.1.5. Gender

Much like in the entire patient population, preoperative CRP positively correlated with postoperative CRP values in male patients on days 1 to 5 and on day 8 as well as in female patients on days 1, 3, 4 and 5 (Table 3; Figure 3). In contrast to the total patient population, leukocyte counts weakly and positively correlated with preoperative CRP values on postoperative days 1 to 3 and on day 5 in male patients, whereas no such positive correlation was seen in female patients. Interestingly, a weak negative correlation of preoperative CRP and postoperative body temperature was seen on days 1 (*r* = −0.273, *n* = 61, *p* = 0.033) and 2 (*r* = −0.248, *n* = 67, *p* = 0.043) after surgery, exclusively in female patients.

#### 3.1.6. Statin Medication

Next, patients were stratified according to treatment with statins and the characteristics of these subgroups are summarized in Table 4. Preoperative and postoperative CRP values positively correlated among patients with or without statin medication during most postoperative days investigated in this study (Table 5; Figure 4a,b). No correlation between preoperative CRP levels and postoperative leukocyte counts was seen, irrespective of stain intake (Table 5; Figure 4c,d). Interestingly, in patients who did not take statins, preoperative CRP and postoperative temperature negatively correlated on days 1 (*r* = −0.204, *n* = 99, *p* = 0.043) and 2 (*r* = −0.200, *n* = 113, *p* = 0.034; Table 5, Figure 4e,f).

We next investigated patients according to statin medication and gender, although due to the low n-number in subgroups, these results should be interpreted with care. Again, pre- and postoperative CRP values positively correlated in men, irrespective of statin intake (Table 6; Figure 5a,b). Statistical significance was reached up to postoperative day 3 (Table 6). In female patients who did not take statins, the positive correlation between pre- and postoperative CRP levels was lost on day two after surgery (Table 7, Figure 6b). In men taking statins, the gender-specific positive correlations of preoperative CRP and early postoperative leukocyte numbers were even more pronounced (*r* = 0.319, *n* = 57, *p* = 0.016) but were absent from men without statin medication (Table 6; Figure 5c,d). In women, no such correlation was seen, irrespective of statin-intake medication (Table 7; Figure 6c,d). The correlation analyses regarding body temperature and preoperative CRP stratification according to statin intake and gender revealed a positive correlation in male statin-taking patients on postoperative day 2 (*r* = 0.327, *n* = 46, *p* = 0.027; Table 6, Figure 5f). The negative correlation of preoperative CRP and early postoperative body temperature typical for women, however, was lost in female patients taking statins and more pronounced in women without this medication on days 1 (*r* = −0.423, *n* = 40, *p* = 0.007) and 2 (*r* = −0.440, *n* = 44, *p* = 0.003) after surgery (Table 7; Figure 6e,f).

### 3.2. CRP-Mediated Control of IL-1β Release by Monocytes from Coronary Heart Disease Patients Treated with Lipid-Lowering Medication

#### 3.2.1. Patient Characteristics

The differences between the surgical patient cohorts with or without statin treatments prompted us to hypothesize that statins impair the anti-inflammatory activity of CRP. To test for this hypothesis, we prospectively investigated statin-naïve patients suffering from low-grade coronary heart disease before intake of statins, 4 weeks after monotherapy with atorvastatin and another 4 weeks after combined treatment with atorvastatin and ASA.

Upon written informed consent, 142 patients admitted to our hospital for a coronary angiography were evaluated for inclusion into this study. Three of them were excluded because they did not suffer from coronary heart disease and another 14 patients because of moderate or severe disease; 103 patients were excluded because of an intake of statins, anticoagulants, antibiotics or immunosuppressants and one patient was excluded because of ASA-intolerance. Finally, 21 patients were included in this study (Figure 7). However, six patients discontinued their participation before the second blood sampling and two before the third sampling (withdrawal of the consent or non-compliance). In total, 13 patients went through the entire study protocol (Figure 7). The characteristics of these patients are summarized in Table 8.

#### 3.2.2. Lipid Levels

As expected, patients treated with atorvastatin alone or in combination with ASA had reduced serum cholesterol and low density lipoprotein (LDL) cholesterol levels but no reduced high density lipoprotein (HDL) cholesterol levels and triglycerides (Appendix A).

#### 3.2.3. Control of ATP-Induced IL-1β Release

To test if lipid-lowering medication impairs the CRP-mediated control of IL-1β release, monocytes were isolated from patient blood, primed with LPS (5 ng/mL, 3 h) and stimulated with BzATP (100 µM, 30 min), a specific agonist of the P2X7 receptor. LPS-primed monocytes released IL-1β in response to BzATP. Of note, the amount of IL-1β released by patient monocytes ex vivo in response to stimulation in vitro was not changed upon lipid-lowering medication (Figure 8).

As described before for peripheral blood mononuclear cells from healthy volunteers [28,29], CRP and other agonists of nicotinic acetylcholine receptors significantly reduced IL-1β release, irrespective of the medication (Appendix A). Only minor changes in LDH concentrations were seen in these experimental settings (Appendix A). Five out of 14 statin-naïve patients were classified as non-responders, as CRP did not reduce the BzATP-induced release of IL-1β below 70%. The data of these patients are depicted in grey in Figure 9 but were not included in the statistical analysis. In line with our hypothesis, the inhibitory effect of CRP (5 µg/mL) on the BzATP induced release of IL-1β was significantly attenuated when patients were treated with atorvastatin and ASA (Figure 9). However, the inhibitory activities of ACh (100 µM), PC (200 µM), nicotine (100 µM), GPC (100 µM) or DPPC (100 µM) on IL-1β release remained by and large unchanged after patient treatment with atorvastatin alone or in combination with ASA (Appendix A).

## 4. Discussion

We set out to test if postoperative CRP levels, leukocyte counts and fever in patients who underwent curative lung cancer surgery correlate with preoperative CRP values. The results of the total patient cohort included in this monocentric, retrospective pilot study essentially revealed neither positive nor negative correlations. However, subgroup analyses with regard to gender and statin intake suggested that in female patients and in those who do not take statins, elevated CRP levels might even correlate with an attenuated trauma-induced increase in inflammatory parameters. In the same line, ex vivo analyses suggested that lipid-lowering medication attenuates the inhibitory effect of CRP on the BzATP-induced release of IL-1β by monocytes.

Patients undergoing thoracic surgery were chosen for the first part of this study, mainly because of two reasons. First, pre-operative CRP is not so much caused by pre-existing inflammatory disease but by tumor-associated factors, such as tumor type and burden. Second, infectious complications play a subordinate role in these patients, at least in the immediate postoperative course. The correlation analyses were based on preoperative CRP levels that were determined during the week preceding surgery rather than on day 0 levels, because the latter were often measured immediately after surgery that might have already caused some changes. The comparison of preoperative and postoperative CRP levels should be interpreted with care because they can be formally regarded as a self-correlation instead of a feedback control system. However, human CRP levels can rise to about 5 mg/L during the first 6 h after an insult and can further increase within the following 48 h to levels up to 500 mg/L [13,35]. The half-life of CRP is about 19 h and its clearance is independent of its concentration [13]. Hence, considerable changes in CRP levels are possible within one or two days. However, the causes for high preoperative CRP levels might be manifold and are not necessarily eliminated by the surgical intervention. A positive correlation of pre- and postoperative CRP levels was seen in the total patient population at all days investigated, and correlation coefficients decreased during the time course. This decrease might be due to the elimination of tumors inducing CRP synthesis. Furthermore, in subgroup analyses regarding gender, smoking behavior, surgical access and extent of pulmonary resection, mainly positive correlations between pre- and postoperative CRP levels were seen.

In sharp contrast, early postoperative (day 1 and day 2) leukocyte counts and fever, two unequivocal hallmarks of inflammation, did not correlate with preoperative CRP levels in the total patient population. Only on days 3 to 5, a very weak positive correlation was seen for leukocyte counts. From the classical point of view, these observations are surprising, because surgical trauma is expected to enhance pre-existing systemic inflammation. In a recent prospective study on patients suffering from multiple traumata, we observed similar negative associations between CRP levels at admission to the hospital and IL-1β levels during the first two days thereafter [29]. Without a doubt, patients with systemic inflammation are at higher risk to develop postoperative complications. Hence, the absence of a positive correlation between preoperative CRP levels and other postoperative markers of inflammation rather favors the hypothesis that CRP exerts anti-inflammatory functions.

Regarding the smoking behavior of the patients, the available data were ambiguous as non-smokers were not explicitly indicated in our documentation system. We can only state that “smokers” were indeed smoking and that “ex-smokers” at least temporarily quit smoking. Non-smokers are included in the mixed patient group for whom we are lacking information. We are interested in a possible effect of smoking because the CRP-dependent inhibition of ATP-induced IL-1β release by human monocytic cells involves nicotinic acetylcholine receptor subunits α7, α9 and α10 [29]. The expression of subunit α7 was induced in regular smokers and in a monocytic cell line treated with nicotine [34]. At the same time, the inhibitory effect of nicotine on LPS-stimulated release of TNF-α was reduced [34]. It is unknown if this also applies to the effects of CRP and nicotinic receptors containing subunits α9 and α10. The observed positive correlation of preoperative CRP levels and leukocyte counts on postoperative day two in ex-smokers might be explained by an impaired responsiveness to CRP. We think that this analysis should be repeated on a larger cohort of patients with a better documentation of self-reported smoking behavior and its verification by measurement of cotinine blood concentrations.

If one considers the surgical access and the extent of pulmonary resection, it is obvious that the number of patients undergoing the most invasive surgery, namely thoracotomy, conversion from VATS to thoracotomy and/or pneumonectomy was too small. However, the data may allow for the hypothesis that the potential anti-inflammatory effects of elevated CRP levels are more pronounced in patients experiencing more severe surgical trauma, as higher levels of cytokines have been measured in response to conventional surgery compared to VATS [36]. Mechanistically, we would expect a higher release of ATP during invasive surgery and, hence, a more pronounced dampening effect of increased CRP levels on the ATP-induced release of cytokines.

Subgroup analysis revealed differences among female and male patients. In male patients, preoperative CRP values and leukocyte counts positively correlated on days 1 to 3 and on day 5, whereas such correlations were seen neither in women nor in the total patient population. In addition, there was a weak negative correlation of preoperative CRP values and early postoperative fever in females on postoperative days 1 and 2 but not in male patients. With all due caution, we speculate that the anti-inflammatory effects of CRP are more pronounced in women. It is known that gender-specific differences in hormones and genetic factors regulate innate immunity, including expression of pattern-recognition receptors, phagocytic activity and antigen presentation [37,38,39,40]. It is conceivable that this also holds true for mechanisms controlling the ATP-induced release of IL-1β. Women have a better outcome after trauma and hemorrhage, a finding that is mainly attributed to protective effects of 17β-estradiol [41] and immunosuppressive effects of androgens [38]. In our study, however, the median age of the patients was 66 years and less than 3% of them were younger than 50 years. At this age, endogenous estradiol levels are low and continuously decrease in women, and testosterone levels are negligible. Unfortunately, hormonal replacement therapies were not documented in our medical records. In men, testosterone levels are reduced by about 50% in comparison to young men, but estradiol levels, although low, can exceed those of women at the same age [39]. Hence, the most pronounced difference in sex hormone levels at this age is related to testosterone. In contrast to hormonal factors, genetic differences are independent of patient age. Several X-linked genes involved in activating pathways of immunity may escape from X chromosome inactivation and result in a higher expression in female compared to male humans [40]. The reasons for the observed gender differences deserve further investigation.

Stratification of patients regarding statin intake revealed a positive correlation of preoperative CRP levels and early postoperative leukocyte counts only in male patients treated with statins. In contrast, only in women who did not take statins, a negative correlation of preoperative CRP and fever was observed. A similar but weaker correlation was seen, as already discussed, in the total population of female patients. These data suggest that anti-inflammatory effects of CRP might be impaired by lipid-lowering medication.

To further test the hypothesis that lipid-lowering medication interferes with anti-inflammatory effects of CRP, the BZATP-induced release of IL-1β by LPS-primed monocytes isolated from patients suffering from cardiovascular disease was investigated in the second part of this study. Monocytes were obtained from patients with newly diagnosed low-grade coronary heart disease before treatment with statins, after 4 weeks of treatment with atorvastatin and after another 4 weeks of treatment with atorvastatin in combination with ASA. The observed reduction in free cholesterol and LDL-cholesterol levels suggested that patients adhered to their medication. BzATP-induced IL-1β release remained unchanged in response to treatment with statins or statins combined with ASA in these ex vivo experiments. These results are in contrast to the well documented anti-inflammatory effects of ASA [42]. Regarding statins, general anti-inflammatory effects have been documented but also an activation of the NLRP3 inflammasome and an increased release of IL-1β [43]. We cannot decide if the effects of ASA and atorvastatin compensated each other in our experimental setting or if they were not active ex vivo.

However, in line with the data obtained from surgical patients, the inhibitory capacity of CRP on the BzATP-induced release of IL-1β was attenuated in monocytes from patients treated with atorvastatin/ASA. These results are in line with the hypothesis that lipid-lowering medications are responsible for an impairment of an anti-inflammatory function of CRP in vivo. However, this coincidence certainly does not prove causality. At present, we cannot explain these observations mechanistically. Furthermore, we ignore why the effects of acetylcholine phosphocholine, nicotine, glycerophosphocholine or dipalmitoyl phosphocholine, which inhibit IL-1β release via mechanisms similar to those of CRP [24,25,26,27], were unimpaired in monocytes isolated from patients treated with atorvastatin and ASA.

This study has numerous inherent limitations because the first part is a retrospective and single-center study that includes a relatively low number of patients, notably in patient subgroups. Only CRP, leukocyte number and body temperature were available to estimate systemic inflammation. We cannot decide if potential anti-inflammatory effects in certain patient subgroups are protective because the clinical outcome was not studied. On the one hand, it may be that reduced inflammatory markers are associated with a beneficial reduction in postoperative SIRS. On the other hand, reduced perioperative inflammation might increase the infection rate and compromise the clinical outcome. More studies are warranted to explain the observed gender-specific effects mechanistically. Regarding the effects of lipid-lowering medicaments on the CRP-mediated inhibition of BzATP-induced release of IL-1β, which were investigated in the second part, we cannot fully differentiate between the underlying disease and the effects of atorvastatin/ASA and, again, larger studies are needed. Furthermore, we excluded the data of all patients from the statistical evaluation, in whom the initial BzATP-induced release of IL-1β was not reduced below 70%, a threshold that was chosen arbitrarily. In addition, it would be of high interest to elucidate the underlying molecular mechanisms. Furthermore, the statistical analyses are obscured by multiple testing and some degree of self-correlation regarding postoperative CRP levels. Therefore, this study should only be considered as a pilot study meant for creating hypotheses.

The following hypotheses warrant further carefully designed prospective multi-center studies. (i) Preoperative CRP levels do not correlate with postoperative inflammation. (ii) High preoperative CRP levels exert anti-inflammatory effects mainly in female patients. (iii) Anti-inflammatory effects of CRP are sensitive to lipid-lowering medication.

After all, who should be afraid of CRP? CRP seems to have at least a dual function in the context of lung cancer surgery. On the one hand, it seems to exacerbate inflammation and tissue damage in organs in the context of surgical trauma and re-ventilation injury. On the other hand, there are increasing data, including data from our group, that suggest that circulating CRP reduces the release of IL-1β in the context of sterile systemic inflammation caused by accidental trauma or surgery. The question of whether CRP exerts pro- or anti-inflammatory functions will probably depend on circumstances: in patients suffering from myocardial infarction and in male surgical patients treated with statins, CRP might fuel inflammation, whereas in female and statin-naïve patients undergoing invasive surgery, high concentrations of endogenous CRP might attenuate inflammation.

## Figures and Tables

**Figure 1 jcm-09-03340-f001:**
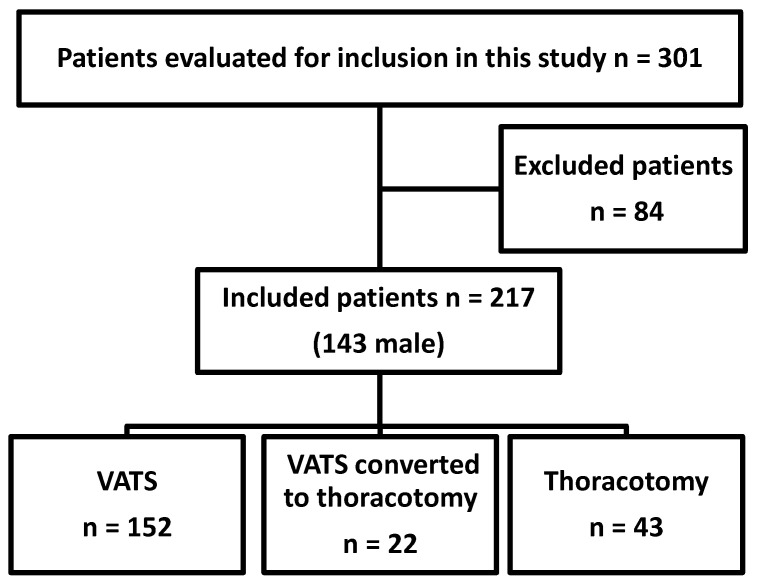
Flow chart of patients undergoing pulmonary resection included in this study stratified for surgical access; VATS, video-assisted thoracic surgery.

**Figure 2 jcm-09-03340-f002:**
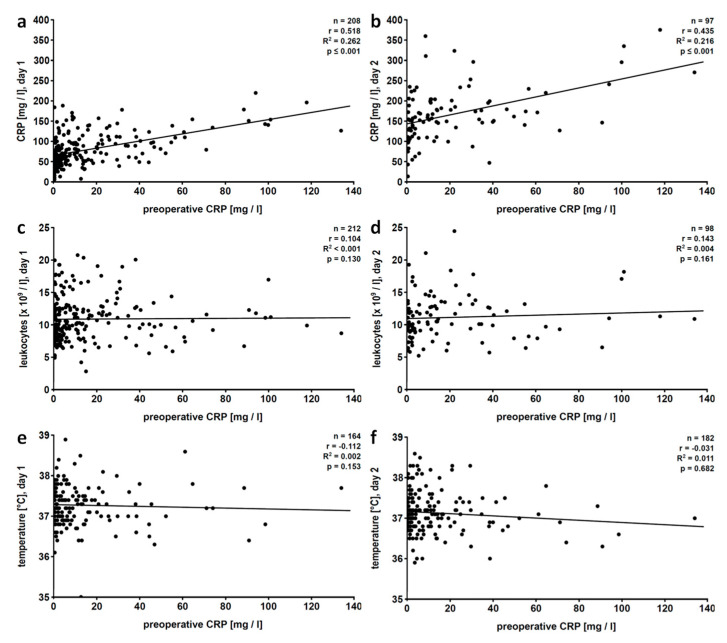
Total patient population: linear regression analyses of preoperative levels of C-reactive protein (CRP) with postoperative CRP levels (**a**,**b**), blood leukocyte counts (**c**,**d**) or fever (**e**,**f**). Patients who underwent pulmonary resection were investigated retrospectively and data collected on the first (**a**,**c**,**e**) and the second (**b**,**d**,**f**) postoperative day are depicted. Preoperative CRP values were obtained within 7 days before surgery. r, correlation coefficient; R^2^, coefficient of determination.

**Figure 3 jcm-09-03340-f003:**
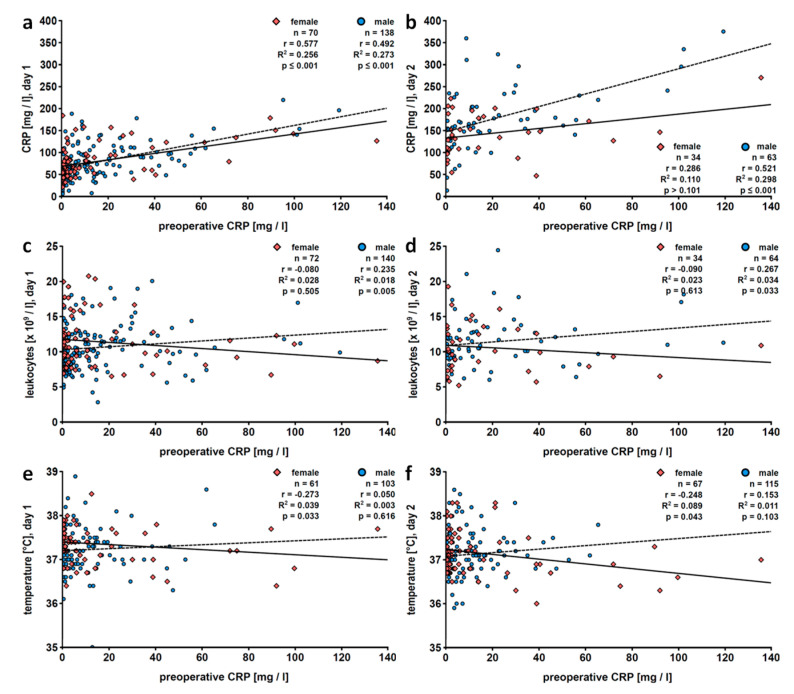
Patients stratified according to gender: linear regression analyses of preoperative levels of C-reactive protein (CRP) with postoperative CRP levels (**a**,**b**), blood leukocyte counts (**c**,**d**) or fever (**e**,**f**). Patients who underwent pulmonary resection were investigated retrospectively and data collected on the first (**a**,**c**,**e**) and the second (**b**,**d**,**f**) postoperative day are depicted. Preoperative CRP values were obtained within 7 days before surgery. The solid line represents the best fit line for data of female patients and the dashed line for those of male patients. r, correlation coefficient; R^2^, coefficient of determination.

**Figure 4 jcm-09-03340-f004:**
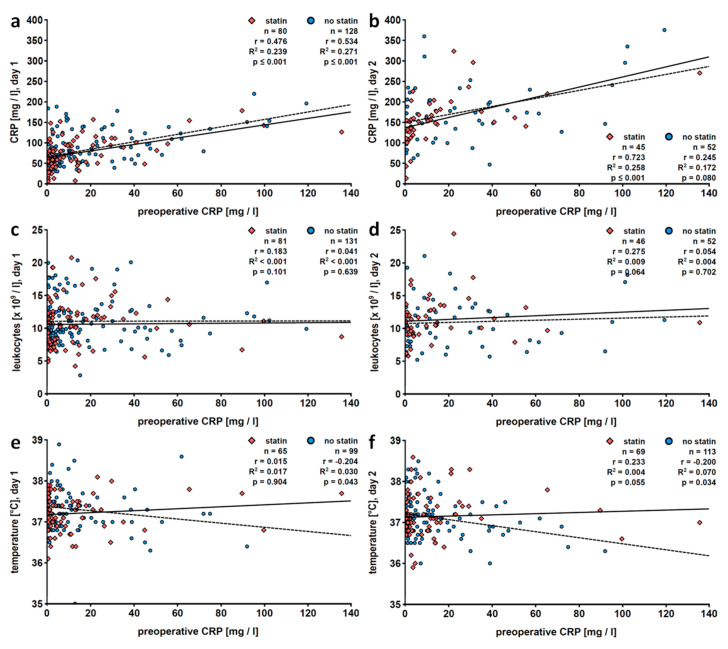
Patients stratified according to statin intake: linear regression analyses of preoperative levels of C-reactive protein (CRP) with postoperative CRP levels (**a**,**b**), blood leukocyte counts (**c**,**d**) or fever (**e**,**f**). Patients who underwent pulmonary resection were investigated retrospectively and data collected on the first (**a**,**c**,**e**) and the second (**b**,**d**,**f**) postoperative day are depicted. Preoperative CRP values were obtained within 7 days before surgery. The solid line represents the best fit line for data patients treated with statins and the dashed line for those of patients who do not take statins. r, correlation coefficient; R^2^, coefficient of determination.

**Figure 5 jcm-09-03340-f005:**
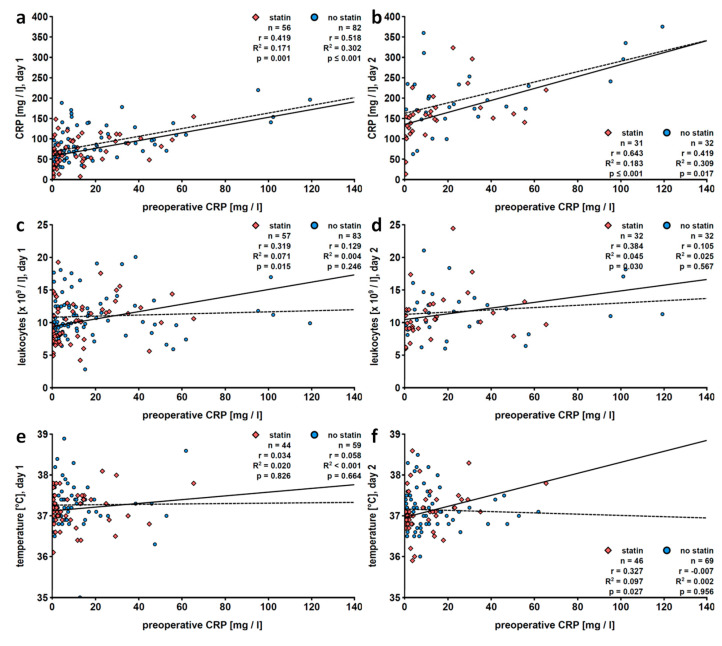
Male patients stratified according to statin intake: linear regression analyses of preoperative levels of C-reactive protein (CRP) with postoperative CRP levels (**a**,**b**), blood leukocyte counts (**c**,**d**) or fever (**e**,**f**). Patients who underwent pulmonary resection were investigated retrospectively and data collected on the first (**a**,**c**,**e**) and the second (**b**,**d**,**f**) postoperative day are depicted. Preoperative CRP values were obtained within 7 days before surgery. The solid line represents the best fit line for data patients treated with statins and the dashed line for those of patients who do not take statins. r, correlation coefficient; R^2^, coefficient of determination.

**Figure 6 jcm-09-03340-f006:**
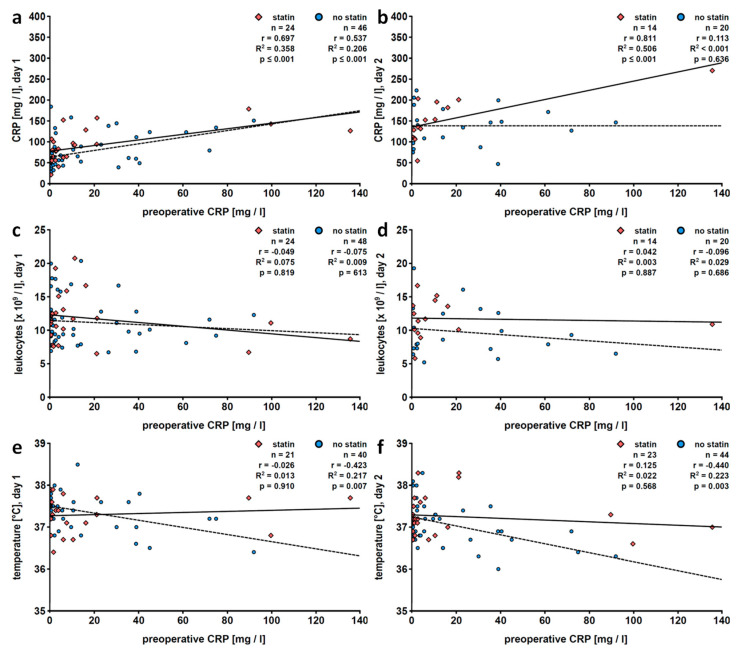
Female patients stratified according to statin intake: linear regression analyses of preoperative levels of C-reactive protein (CRP) with postoperative CRP levels (**a**,**b**), blood leukocyte counts (**c**,**d**) or fever (**e**,**f**). Patients who underwent pulmonary resection were investigated retrospectively and data collected on the first (**a**,**c**,**e**) and the second (**b**,**d**,**f**) postoperative day are depicted. Preoperative CRP values were obtained within 7 days before surgery. The solid line represents the best fit line for data patients treated with statins and the dashed line for those of patients who do not take statins. r, correlation coefficient; R^2^, coefficient of determination.

**Figure 7 jcm-09-03340-f007:**
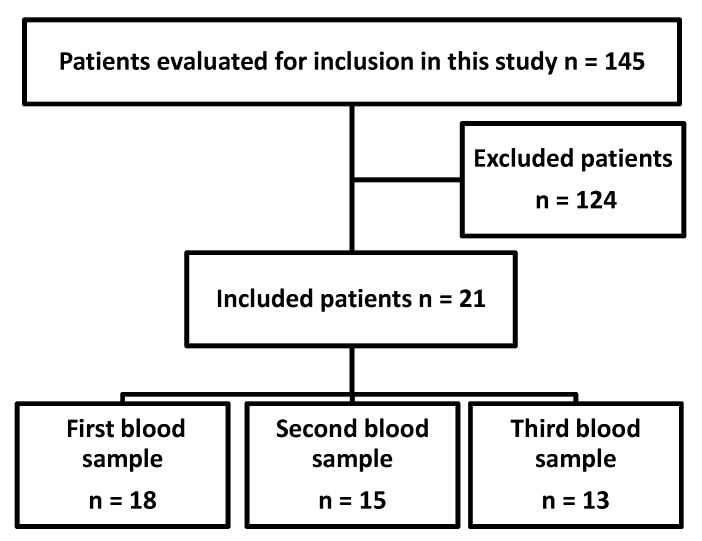
Flow chart of patients undergoing coronary angiography included in this study.

**Figure 8 jcm-09-03340-f008:**
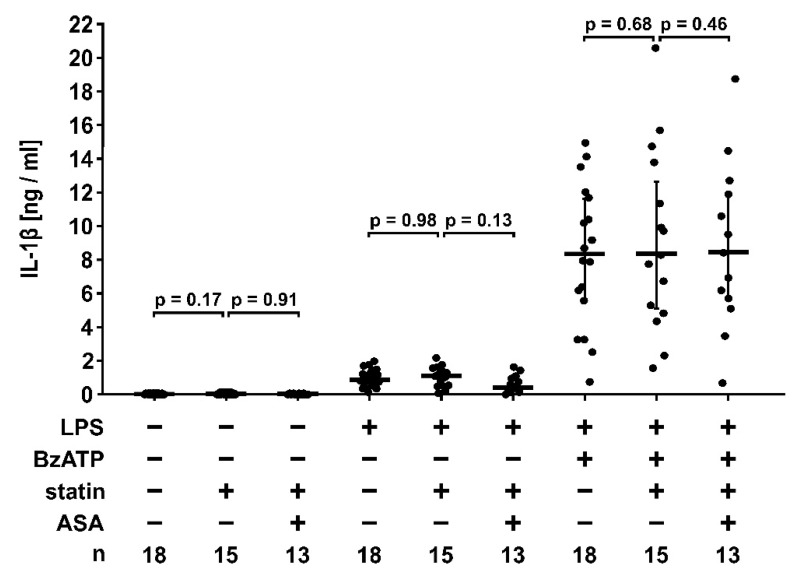
Release of interleukin-1β (IL-1β) by monocytes isolated from the blood of patients suffering from low-grade coronary heart disease. Patient blood was drawn before lipid-lowering medication, after four weeks of statin intake and after another four weeks of combined treatment with statin and acetylsalicylic acid (ASA). IL-1β was measured in cell culture supernatants of unstimulated cells, cells primed with lipopolysaccharide (LPS, 5 ng/mL, 3 h) or LPS-primed cells stimulated with 2′(3′)-*O*-(4-benzoylbenzoyl)adenosine 5′-triphosphate (BzATP, 100 µM, 30 min). Data are presented as individual data points; bar represents median; whiskers encompass the 25th to 75th percentile. Experimental groups were compared by the Friedman test followed by the Wilcoxon signed-rank test.

**Figure 9 jcm-09-03340-f009:**
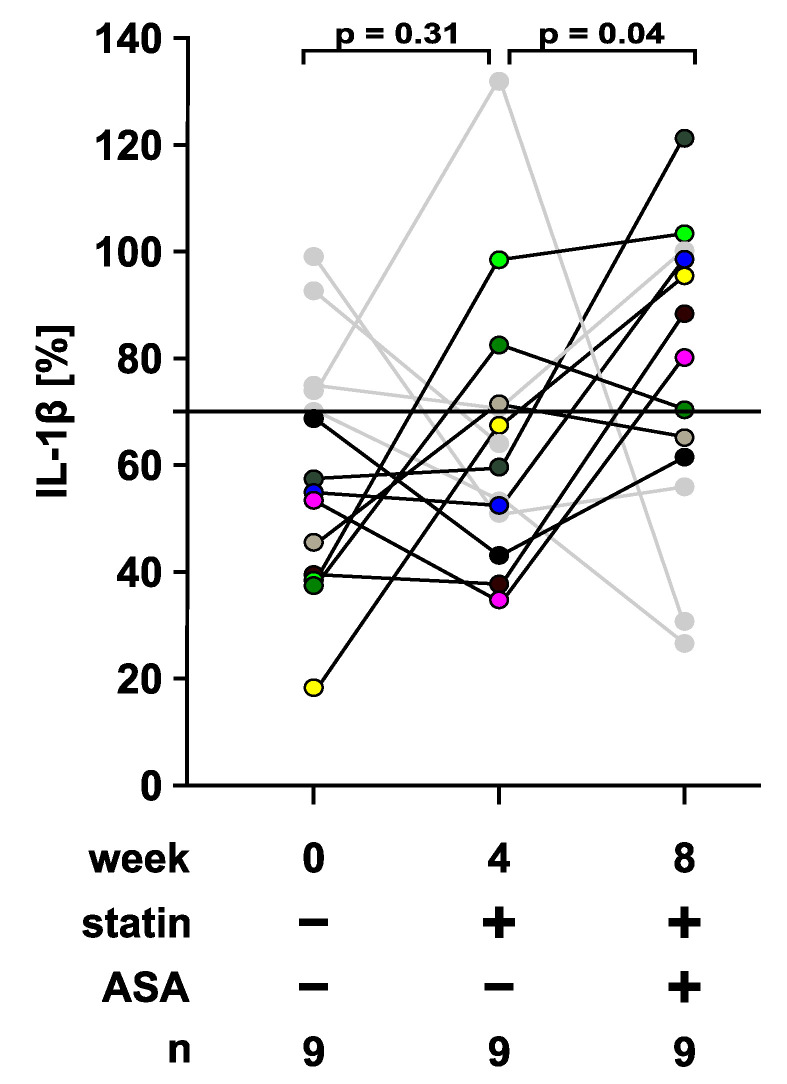
C-reactive protein (CRP)-mediated inhibition of the release of interleukin-1β (IL-1β) by monocytes of patients suffering from low-grade coronary heart disease. Patient blood was drawn before lipid-lowering medication, after four weeks of statin intake and after another four weeks of combined treatment with statin and acetylsalicylic acid (ASA). IL-1β was measured in cell culture supernatants of monocytes primed with lipopolysaccharide (LPS, 5 ng/mL, 3 h) and stimulated with 2′(3′)-*O*-(4-benzoylbenzoyl)adenosine 5′-triphosphate (BzATP, 100 µM, 30 min) in the presence or absence of CRP (5 µg/mL). IL-1β values obtained upon stimulation with BzATP were set to 100% and the values obtained in the presence of CRP were calculated accordingly. Data from patients who did not show a CRP-mediated inhibition of IL-1β release below 70% are depicted in grey and were not included in the statistical analysis. Data are presented as individual data points; data from individual patients are symbolized by different colors; data points belonging to the same individual are connected by lines; bar represents median; whiskers encompass the 25th to 75th percentile. Experimental groups were compared by Friedman test followed by the Wilcoxon signed-rank test.

**Table 1 jcm-09-03340-t001:** Characteristics of pulmonary cancer patients included in this study stratified according to gender.

Patient Characteristics	Male	Female	*p*	All Patients
Age (median, interquartile ranges, (years))	67 (60–74)	66 (59–74)	0.346	66 (60–74)
Sex	143 (66%)	74 (34%)		217
Smoking behavior				
smoker	31 (22%)	23 (31%)	0.295	54 (25%)
ex-smoker	34 (24%)	14 (19%)		48 (22%)
unknown	78 (54%)	37 (50%)		115 (53%)
Preoperative CRP (median, interquartile range, (mg/L))	7.8 (2.1–20.4)	3.7 (0.8–17.5)	**0.04**	6.1 (1.5–20)
Preoperative leukocyte number (median, interquartile range, (×10^3^/µL))	7.8 (6.7–9.0)	8 (6.9–9.5)	0.483	7.8 (6.8–9.2)
Histopathology				
adenocarcinoma	64 (45%)	46 (62%)		110 (51%)
squamous cell carcinoma	61 (43%)	16 (22%)	**0.010**	77 (36%)
SCLC	11 (8%)	4 (5%)	0.096	15 (7%)
carcinoid tumor	4 (2%)	6 (8%)		10 (5%)
other	3 (2%)	2 (3%)		5 (2%)
Tumor stages *				
Ia	37 (26%)	26 (36%)		63 (29%)
Ib	25 (17%)	17 (23%)		42 (19%)
IIa	27 (19%)	6 (8%)		33 (15%)
IIb	24 (17%)	8 (11%)		32 (15%)
III	26 (18%)	13 (18%)		39 (18%)
IV	4 (3%)	3 (4%)		7 (3%)
Surgical access				
VATS	103 (72%)	49 (66%)	0.646	152 (70%)
conversion from VATS to TT	13 (9%)	9 (12%)		22 (10%)
TT	27 (19%)	16 (22%)		43 (20%)
Extent of pulmonary resection				
segmentectomy	10 (7%)	5 (7%)	0.330	15 (7%)
lobectomy	126 (88%)	69 (93%)		195 (90%)
pneumonectomy	7 (5%)	0 (0%)		7 (3%)
Postoperative inflammatory complications	65 (46%)	29 (39%)	0.377	94 (43%)

Age, preoperative CRP and preoperative leukocyte number were analyzed using the Mann–Whitney U test and all other parameters using the chi-squared test. * Data for one female patient were missing. SCLC, small cell lung cancer; TT, thoracotomy; VATS, video-assisted thoracoscopic surgery. The bold is a common way to highlight *p*-values below 5%.

**Table 2 jcm-09-03340-t002:** Linear regression analysis of preoperative C-reactive protein (CRP) values versus postoperative CRP, leukocyte number and body temperature (*n* = 217).

Day	Postoperative CRP	Leukocyte Number	Temperature
0	0.868 (*n* = 180, *p* ≤ 0.001)	0.010 (*n* = 205)	0.118 (*n* = 150)
1	0.518 (*n* = 208, *p* ≤ 0.001)	0.104 (*n* = 212)	−0.112 (*n* = 164)
2	0.435 (*n* = 97, *p* ≤ 0.001)	0.143 (*n* = 98)	−0.031 (*n* = 182)
3	0.531 (*n* = 115, *p* ≤ 0.001)	0.215 (*n* = 115, *p* = 0.021)	0.013 (*n* = 187)
4	0.397 (*n* = 126, *p* ≤ 0.001)	0.199 (*n* = 126, *p* = 0.025)	0.072 (*n* = 185)
5	0.367 (*n* = 96, *p* ≤ 0.001)	0.203 (*n* = 97, *p* = 0.046)	0.013 (*n* = 182)
6	0.314 (*n* = 86, *p* = 0.003)	0.064 (*n* = 87)	0.018 (*n* = 168)
7	0.293 (*n* = 9, *p* = 0.005)	0.118 (*n* = 90)	−0.014 (*n* = 157)
8	0.315 (*n* = 67, *p* = 0.009)	0.119 (*n* = 66)	0.014 (*n* = 129)
9	0.262 (*n* = 59, *p* = 0.045)	0.132 (*n* = 60)	0.047 (*n* = 113)
10	0.259 (*n* = 53, *p* ≤ 0.001)	0.183 (*n* = 53)	0.083 (*n* = 97)

All patients included in this study are analyzed. The correlation coefficients (*r*) as well as *n*-numbers and *p*-values (≤ 0.05) are given.

**Table 3 jcm-09-03340-t003:** Gender-specific subgroup analysis (male, *n* = 143; female, *n* = 74).

Day	Postoperative CRP, Male	Postoperative CRP, Female
0	0.851 (*n* = 121, *p* ≤ 0.001)	0.864 (*n* = 59, *p* ≤ 0.001)
1	0.492 (*n* = 138, *p* ≤ 0.001)	0.577 (*n* = 70, *p* ≤ 0.001)
2	0.521 (*n* = 63, *p* ≤ 0.001)	0.286 (*n* = 34)
3	0.534 (*n* = 77, *p* ≤ 0.001)	0.463 (*n* = 38, *p* = 0.003)
4	0.263 (*n* = 80, *p* = 0.018)	0.480 (*n* = 46, *p* ≤ 0.001)
5	0.265 (*n* = 61, *p* = 0.039)	0.485 (*n* = 35, *p* = 0.003)
6	0.214 (*n* = 63)	0.378 (*n* = 23)
7	0.217 (*n* = 57)	0.320 (*n* = 34)
8	0.291 (*n* = 50, *p* = 0.040)	0.262 (*n* = 17)
9	0.212 (*n* = 45)	0.200 (*n* = 14)
10	0.245 (*n* = 34)	0.246 (*n* = 19)
**Day**	**Leuko, Male**	**Leuko, Female**
0	0.047 (*n* = 135)	−0.029 (*n* = 70)
1	0.235 (*n* = 140, *p* = 0.005)	−0.080 (*n* = 72)
2	0.267 (*n* = 64, *p* = 0.033)	−0.090 (*n* = 34)
3	0.356 (*n* = 77, *p* ≤ 0.001)	−0.028 (*n* = 38)
4	0.193 (*n* = 80)	0.160 (*n* = 46)
5	0.375 (*n* = 62, *p* = 0.003)	−0.081 (*n* = 35)
6	0.067 (*n* = 64)	−0.017 (*n* = 23)
7	0.095 (*n* = 56)	0.182 (*n* = 34)
8	0.193 (*n* = 49)	−0.130 (*n* = 17)
9	0.073 (*n* = 46)	0.510 (*n* = 14)
10	0.092 (*n* = 34)	0.335 (*n* = 19)
**Day**	**Temp, Male**	**Temp, Female**
0	0.094 (*n* = 100)	0.236 (*n* = 50)
1	0.050 (*n* = 103)	−0.273 (*n* = 61, *p* = 0.033)
2	0.153 (*n* = 115)	−0.248 (*n* = 67, *p* = 0.043)
3	0.047 (*n* = 119)	−0.014 (*n* = 68)
4	0.077 (*n* = 117)	0.126 (*n* = 68)
5	0.125 (*n* = 111)	−0.045 (*n* = 71)
6	−0.031 (*n* = 105)	0.110 (*n* = 63)
7	−0.025 (*n* = 100)	0.025 (*n* = 57)
8	0.134 (*n* = 82)	−0.155 (*n* = 47)
9	0.128 (*n* = 72)	−0.033 (*n* = 41)
10	0.195 (*n* = 59)	−0.036 (*n* = 38)

Linear regression analysis of preoperative C-reactive protein (CRP) values versus postoperative CRP, leukocyte number (leuko) and body temperature (temp). The correlation coefficients (*r*) as well as *n*-numbers and *p*-values (≤0.05) are given.

**Table 4 jcm-09-03340-t004:** Characteristics of patients (*n* = 217) included in this study stratified according to statin intake.

Patient Characteristics	Statin (*n* = 86)	No Statin (*n* = 131)	*p*	All Patients
Age (median, interquartile range, (years))	71 (65–75)	64 (58–72)	**≤0.001**	66 (60–74)
Sex	86 (39.6%)	131 (60.4%)		217
Smoking behavior				
smoker	15 (17.4%)	39 (29.8%)		54 (25%)
ex-smoker	20 (23.3%)	28 (21.4%)	0.115	48 (22%)
unknown	51 (59.3%)	64 (48.9%)		115 (53%)
Preoperative CRP (median, interquartile range, (mg/L))	3.9 (1.2–14.6)	7.4 (1.6–22.8)	0.173	6.1 (1.5–19.9)
Preoperative leukocyte number (median, interquartile range (×10^3^/µL))	8.0 (6.9–8.8)	7.8 (6.6–9.5)	0.818	7.8 (6.8–9.2)
Histopathology				
adenocarcinoma	41 (47.7%)	69 (52.7%)		110 (51%)
squamous cell carcinoma	33 (38.4%)	44 (33.6%)	0.744	77 (36%)
SCLC	9 (10.5%)	6 (4.6%)		15 (7%)
carcinoid tumor	1 (1.2%)	9 (6.9%)	**0.012**	10 (5%)
other	2 (2.3%)	3 (2.3%)		5 (2%)
Surgical access				
VATS	62 (72.1%)	90 (68.7%)		152 (70%)
conversion from VATS to TT	11 (12.8%)	11 (8.4%)	0.264	22 (10%)
TT	13 (15.1%)	30 (22.9%)		43 (20%)
Extent of pulmonary resection				
segmentectomy	7 (8.1%)	8 (6.1%)		15 (7%)
lobectomy	78 (90.7%)	117 (89.3%)	0.333	195 (90%)
pneumonectomy	1 (1.3%)	6 (4.6%)		7 (3%)
Perioperative (d0–d10) inflammatory complications	37 (43%)	57 (43.5%)	0.943	94 (43%)

Age, preoperative CRP and preoperative leukocyte number were analyzed using the Mann–Whitney U test and all other parameters using the chi-squared test. SCLC, small cell lung cancer; TT, thoracotomy; VATS, video-assisted thoracoscopic surgery. The bold is a common way to highlight *p*-values below 5%.

**Table 5 jcm-09-03340-t005:** Subgroup correlation analysis regarding statin medication.

Day	Postoperative CRP, Statin	Postoperative CRP, No Statin
0	0.828 (*n* = 69, *p* ≤ 0.001)	0.886 (*n* = 111, *p* ≤ 0.001)
1	0.476 (*n* = 80, *p* ≤ 0.001)	0.534 (*n* = 128, *p* ≤ 0.001)
2	0.723 (*n* = 45, *p* ≤ 0.001)	0.245 (*n* = 52)
3	0.700 (*n* = 52, *p* ≤ 0.001)	0.416 (*n* = 63, *p* ≤ 0.001)
4	0.469 (*n* = 5, *p* ≤ 0.001)	0.337 (*n* = 74, *p* = 0.003)
5	0.380 (*n* = 96, *p* = 0.016)	0.394 (*n* = 56, *p* = 0.003)
6	0.169 (*n* = 36)	0.438 (*n* = 50, *p* ≤ 0.001)
7	0.328 (*n* = 91, *p* = 0.030)	0.398 (*n* = 47, *p* = 0.006)
8	0.206 (*n* = 25)	0.401 (*n* = 42, *p* = 0.009)
9	0.323 (*n* = 24)	0.311 (*n* = 35)
10	0.378 (*n* = 22)	0.273 (*n* = 31)
**Day**	**Leuko, Statin**	**Leuko, No Statin**
0	0.054 (*n* = 82)	−0.039 (*n* = 123)
1	0.183 (*n* = 81)	0.041 (*n* = 131)
2	0.275 (*n* = 46)	0.054 (*n* = 52)
3	0.451(*n* = 51, *p* ≤ 0.001)	0.045 (*n* = 64)
4	0.114 (*n* = 52)	0.263 (*n* = 74, *p* = 0.023)
5	0.158 (*n* = 40)	0.255 (*n* = 57)
6	0.067 (*n* = 37)	0.109 (*n* = 50)
7	0.143 (*n* = 44)	0.103 (*n* = 46)
8	0.178 (*n* = 25)	0.105 (*n* = 41)
9	0.096 (*n* = 25)	0.203 (*n* = 35)
10	0.569 (*n* = 22, *p* = 0.006)	−0.023 (*n* = 31)
**Day**	**Temp, Statin**	**Temp, No Statin**
0	−0.042 (*n* = 51)	0.197 (*n* = 99)
1	0.015 (*n* = 65)	−0.204 (*n* = 99, *p* = 0.043)
2	0.233 (*n* = 69)	−0.200 (*n* = 113, *p* = 0.034)
3	0.025 (*n* = 73)	0.006 (*n* = 114)
4	0.181 (*n* = 70)	0.033 (*n* = 115)
5	0.15 (*n* = 69)	−0.069 (*n* = 113)
6	0.034 (*n* = 61)	−0.012 (*n* = 107)
7	0.113 (*n* = 59)	−0.093 (*n* = 98)
8	−0.022 (*n* = 53)	0.038 (*n* = 76)
9	0.027 (*n* = 45)	0.075 (*n* = 68)
10	−0.059 (*n* = 36)	0.204 (*n* = 61)

Linear regression analysis of preoperative CRP values versus postoperative CRP, leukocyte number (leuko) and body temperature (temp). The correlation coefficients (*r*), as well as *n*-numbers and *p*-values (≤0.05) are given.

**Table 6 jcm-09-03340-t006:** Subgroup correlation analysis, male patients regarding statin medication.

Day	Postoperative CRP, Statin	Postoperative CRP, No Statin
0	0.836 (*n* = 47, *p* ≤ 0.001)	0.849 (*n* = 74, *p* ≤ 0.001)
1	0.419 (*n* = 56, *p* ≤ 0.001)	0.518 (*n* = 82, *p* ≤ 0.001)
2	0.643 (*n* = 31, *p* ≤ 0.001)	0.419 (*n* = 32, *p* = 0.017)
3	0.717 (*n* = 38, *p* ≤ 0.001)	0.338 (*n* = 39, *p* = 0.035)
4	0.369 (*n* = 34, *p* = 0.032)	0.166 (*n* = 46)
5	0.268 (*n* = 23)	0.288 (*n* = 38)
6	0.262 (*n* = 27)	0.164 (*n* = 36)
7	0.491 (*n* = 28, *p* = 0.008))	0.186 (*n* = 29)
8	0.412 (*n* = 18)	0.284 (*n* = 32)
9	0.295 (*n* = 19)	0.190 (*n* = 26)
10	0.42 (*n* = 12)	0.222 (*n* = 22)
**Day**	**Leuko, Statin**	**Leuko, No Statin**
0	0.149 (*n* = 57)	−0.071 (*n* = 78)
1	0.319 (*n* = 57, *p* = 0.015)	0.129 (*n* = 83)
2	0.384 (*n* = 32, *p* = 0.030)	0.105 (*n* = 32)
3	0.609 (*n* = 38, *p* ≤ 0.001)	0.119 (*n* = 39)
4	0.291 (*n* = 34)	0.148 (*n* = 46)
5	0.434 (*n* = 23, *p* = 0.039)	0.330 (*n* = 39, *p* = 0.041)
6	0.125 (*n* = 28)	0.063 (*n* = 36)
7	0.107 (*n* = 28)	0.111 (*n* = 28)
8	0.326 (*n* = 18)	0.131 (*n* = 31)
9	−0.023 (*n* = 20)	0.247 (*n* = 26)
10	0.650 (*n* = 12, *p* = 0.022)	−0.082 (*n* = 22)
**Day**	**Temp, Statin**	**Temp, No Statin**
0	−0.052 (*n* = 36)	0.168 (*n* = 64)
1	0.034 (*n* = 44)	0.058 (*n* = 59)
2	0.327 (*n* = 46, *p* = 0.027)	−0.007 (*n* = 69)
3	0.114 (*n* = 49)	−0.017 (*n* = 70)
4	0.130 (*n* = 47)	0.072 (*n* = 70)
5	0.244 (*n* = 45)	0.036 (*n* = 66)
6	−0.152 (*n* = 42)	0.022 (*n* = 63)
7	0.205 (*n* = 40)	−0.159 (*n* = 60)
8	0.043 (*n* = 35)	0.232 (*n* = 47)
9	0.05 (*n* = 31)	0.242 (*n* = 41)
10	−0.022 (*n* = 23)	0.398 (*n* = 36, *p* = 0.016)

Linear regression analysis of preoperative CRP values versus postoperative CRP, leukocyte number (leuko) and body temperature (temp). The correlation coefficients (*r*) as well as *n*-numbers and *p*-values (≤0.05) are given.

**Table 7 jcm-09-03340-t007:** Subgroup correlation analysis, female patients regarding statin medication.

Day	Postoperative CRP, Statin	Postoperative CRP, No Statin
0	0.750 (*n* = 22, *p* ≤ 0.001)	0.911 (*n* = 37, *p* ≤ 0.001)
1	0.697 (*n* = 24, *p* ≤ 0.001)	0.537 (*n* = 46, *p* ≤ 0.001)
2	0.811 (*n* = 14, *p* ≤ 0.001)	0.113 (*n* = 20)
3	0.479 (*n* = 14)	0.474 (*n* = 24, *p* = 0.019)
4	0.464 (*n* = 18)	0.463 (*n* = 28, *p* = 0.013)
5	0.550 (*n* = 17, *p* = 0.022)	0.493 (*n* = 18, *p* = 0.038)
6	−0.133 (*n* = 9)	0.572 (*n* = 14, *p* = 0.033)
7	0.071 (*n* = 16)	0.545 (*n* = 18, *p* = 0.019)
8	−0.286 (*n* = 7)	0.576 (*n* = 10)
9	0.300 (*n* = 5)	0.250 (*n* = 9)
10	0.333 (*n* = 10)	0.150 (*n* = 9)
**Day**	**Leuko, Statin**	**Leuko, No Statin**
0	−0.055 (*n* = 25)	0.030 (*n* = 45)
1	−0.049 (*n* = 24)	−0.075 (*n* = 48)
2	0.042 (*n* = 14)	−0.096 (*n* = 20)
3	−0.160 (*n* = 14)	0.018 (*n* = 25)
4	−0.106 (*n* = 18)	0.359 (*n* = 28)
5	−0.231 (*n* = 17)	0.077 (*n* = 18)
6	−0.050 (*n* = 9)	0.024 (*n* = 14)
7	0.193 (*n* = 16)	0.138 (*n* = 18)
8	−0.214 (*n* = 7)	0.006 (*n* = 10)
9	0.400 (*n* = 5)	0.467 (*n* = 9)
10	0.479 (*n* = 10)	0.293 (*n* = 9)
**Day**	**Temp, Statin**	**Temp, No Statin**
0	0.163 (*n* = 15)	0.222 (*n* = 35)
1	−0.026 (*n* = 21)	−0.423 (*n* = 40, *p* = 0.007)
2	0.125 (*n* = 23)	−0.440 (*n* = 44, *p* = 0.003)
3	−0.154 (*n* = 24)	0.059 (*n* = 44)
4	0.292 (*n* = 23)	0.056 (*n* = 45)
5	0.003 (*n* = 24)	−0.078 (*n* = 47)
6	0.430 (*n* = 19)	−0.016 (*n* = 44)
7	−0.074 (*n* = 19)	0.001 (*n* = 38)
8	−0.204 (*n* = 18)	−0.198 (*n* = 29)
9	−0.088 (*n* = 14)	0.040 (*n* = 27)
10	−0.096 (*n* = 13)	−0.023 (*n* = 25)

Linear regression analysis of preoperative CRP values versus postoperative CRP, leukocyte number (leuko) and body temperature (temp). The correlation coefficients (*r*) as well as *n*-numbers and *p*-values (≤0.05) are given.

**Table 8 jcm-09-03340-t008:** Characteristics of patients suffering from coronary heart disease included in this study (data before coronary angiography).

Patient Number	21
Gender	
male	12 (57%)
female	9 (43%)
Age (median, interquartile range, (years))	60 (55–65)
BMI (median, interquartile range, (kg/m^2^))	29 (24.5–32.6)
Smoking behavior	
smoker	6 (29%)
non-smoker	15 (71%)
Leukocytes (median, interquartile ranges, (10^3^/µL))	6.70 (5.70–8.50)
Serum cholesterol (median, interquartile ranges, (mg/dL))	221 (204–247)
Serum LDL cholesterol (median, interquartile ranges, (mg/dL))	139 (126–152)
Serum HDL cholesterol (median, interquartile ranges, (mg/dL))	54 (42.50–69.25)
Serum triglycerides (median, interquartile ranges, (mg/dL))	138 (111–211)
Comorbidities	
arterial hypertension	7 (33%)
COPD	1 (5%)
diabetes type 2	1 (5%)
fibromyalgia	1 (5%)
hyperuricemia	1 (5%)
hypothyroidism	6 (29%)

BMI, body mass index; COPD, chronic obstructive pulmonary disease; HDL, high density lipoprotein; LDL, low density lipoprotein.

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
