# Peer review of "Who Is Afraid of CRP? Elevated Preoperative CRP Levels Might Attenuate the Increase in Inflammatory Parameters in Response to Lung Cancer Surgery"

_jcm, 2020, doi:10.3390/jcm9103340_

Round 1
Reviewer 1 Report
The manuscript has been significantly improved and now can be accepted for publication in JCM.
Author Response
We thank the reviewer for this positive comment.
Reviewer 2 Report
Authors have tried to respond to all me questions.However, I don't think authors have shown sufficient clinical data to conclude that CRP is protective against lung cancer surgery.
Author Response
We are very grateful that you pointed out this important weakness in the first version of our manuscript. You were quite right to state that we do not provide evidence that CRP can be protective in the context of lung surgery. Hence, we amended all parts of the manuscript to avoid this undue conclusion.
For the avoidance of any doubt, we now additionally included the following statement in the limitations of the study in lines 515 to 519: “We cannot decide if potential anti-inflammatory effects in certain patient subgroups are protective because the clinical outcome was not studied. On the one hand, it may be that reduced inflammatory markers are associated with a beneficial reduction of postoperative SIRS. On the other hand, reduced perioperative inflammation might increase the infection rate and compromise the clinical outcome.”
Round 2
Reviewer 2 Report
I think authors have responded to my questions and the conclusion is reasonable in this revised version. I do not have additional questions.
This manuscript is a resubmission of an earlier submission. The following is a list of the peer review reports and author responses from that submission.
Round 1
Reviewer 1 Report
In this study, the authors confirmed if preoperative CRP levels were correlated with postoperative CRP, leukocyte counts, and fever in the context of anatomical lung resection and systematic lymph node dissection as first-line lung cancer therapy. They found that high CRP levels protect from trauma-induced inflammation.
This a well-written manuscript. I have no major concerns.
Reviewer 2 Report
In this paper, the authors attempt to examine relationship between preoperative and postoperative CRP values. This is an interesting research, however I have several questions. I am listing my comments for the authors below.
Introduction is redundant and difficult to follow the main purposes of this study.
The definition of "being protective" is also unclear in this study. Authors should set clinically relevant endpoints if you assert that CRP is protective. For example, correlation CRP trends with postoperative complications would be needed to evaluate if CRP is protective in short-term outcomes of patients undergoing lung surgery.
Table 2 suggests that there are many dropouts (lacking data) in blood samples in this study. Thus, it is quite difficult to say based on this data that CRP can be protective in lung surgery. I am unclear as to how these missing data were treated in the analytical models.
If authors cannot justify the situation, a longitudinal model, such as linear mixed model, that allows you to use include missing data would be needed.
Pathological stage should be added in Table 1 as it can affect CRP values. Is CRP very highly correlated with stage or histology? If it is, are the findings in the model driven largely by this? Would be interesting to see the correlation between stage and CRP levels.
Reviewer 3 Report
In this manuscript, the authors retrospectively analyzed the correlation between the level of C-reactive protein (CRP) and the severity of inflammation in the context of anatomical lung resection and systematic lymph node dissection. They stratified patients based on the smoking behavior, surgical access and extend of pulmonary resection, gender, and statin intake, and then performed the linear regression analysis of preoperative CRP values versus postoperative CRP, leukocyte number and body temperature. They found that the preoperative CRP levels only correlated with postoperative CRPS values, rather than others. Interestingly, they found that in men taking statins, the gender-specific positive correlations of preoperative CRP and early postoperative leukocyte numbers were more pronounced, compared to the man without statin medication. However, the negative correlation of preoperative CRP and early postoperative body temperature is more pronounced in women without this medication, compared to the women taking statins. In addition, they showed that CRP is able to mediate the inhibition of the release of interleukin-1β (IL-1β) in the monocytes of patients suffering from low-grade coronary heart disease. In general, the manuscript is well written. Most of the data present supports the conclusion. My comment is shown as below:
In Fig.4, Fig.5, and Fig.6, the data is not properly interpreted. For instance, the authors didn’t provide any interpretation for Fig.5a, Fig.5b, Fig.5e, Fig.5f. It would be appreciated if the authors would do more digestion.
In table 8, the information of CRP levels in the selected patients should be included.
In Fig.9, is there any correlation between the endogenous CRP levels and the interleukin-1β production in the three groups.
In line 368, I guess “Fife” would be a typo.